# Oncogenic and Tumor Suppressive Components of the Cell Cycle in Breast Cancer Progression and Prognosis

**DOI:** 10.3390/pharmaceutics13040569

**Published:** 2021-04-17

**Authors:** Dharambir Kashyap, Vivek Kumar Garg, Elise N. Sandberg, Neelam Goel, Anupam Bishayee

**Affiliations:** 1Department of Histopathology, Postgraduate Institute of Medical Education and Research, Chandigarh 160 012, Punjab, India; make.must@gmail.com; 2Punjab Biotechnology Incubator, Mohali 160 059, Punjab, India; garg.vivek85@gmail.com; 3Lake Erie College of Osteopathic Medicine, Bradenton, FL 34211, USA; Elise.sandberg@gmail.com; 4University Institute of Engineering and Technology, Panjab University, Chandigarh 160 014, Punjab, India

**Keywords:** cell cycle, cyclin-dependent kinase, p16, p21, p27, breast cancer, prognosis

## Abstract

Cancer, a disease of inappropriate cell proliferation, is strongly interconnected with the cell cycle. All cancers consist of an abnormal accumulation of neoplastic cells, which are propagated toward uncontrolled cell division and proliferation in response to mitogenic signals. Mitogenic stimuli include genetic and epigenetic changes in cell cycle regulatory genes and other genes which regulate the cell cycle. This suggests that multiple, distinct pathways of genetic alterations lead to cancer development. Products of both oncogenes (including cyclin-dependent kinase (CDKs) and cyclins) and tumor suppressor genes (including cyclin-dependent kinase inhibitors) regulate cell cycle machinery and promote or suppress cell cycle progression, respectively. The identification of cyclins and CDKs help to explain and understand the molecular mechanisms of cell cycle machinery. During breast cancer tumorigenesis, cyclins A, B, C, D1, and E; cyclin-dependent kinase (CDKs); and CDK-inhibitor proteins p16, p21, p27, and p53 are known to play significant roles in cell cycle control and are tightly regulated in normal breast epithelial cells. Following mitogenic stimuli, these components are deregulated, which promotes neoplastic transformation of breast epithelial cells. Multiple studies implicate the roles of both types of components—oncogenic CDKs and cyclins, along with tumor-suppressing cyclin-dependent inhibitors—in breast cancer initiation and progression. Numerous clinical studies have confirmed that there is a prognostic significance for screening for these described components, regarding patient outcomes and their responses to therapy. The aim of this review article is to summarize the roles of oncogenic and tumor-suppressive components of the cell cycle in breast cancer progression and prognosis.

## 1. Introduction

Cancer, a disease of uncontrolled cell division, is known to exhibit a series of changes in the activity of cell cycle regulators [1]. All cancer types arise from a single cell that has transformed due to genetic or regulatory alterations, resulting in uncontrolled cell division in response to mitogenic signals [2]. Mitogenic signals include genetic and epigenetic aberrations in cell cycle regulatory genes. These mitogenic stimuli make oncogenic changes, resulting in cell transformation [3]. The gain-of-function mutations cause the activation of proto-oncogenes, which are normally present in the suppressed state in differentiated cells under epigenetic control [4]. Oncogenic stimuli have the potential to induce transformation of the differentiated cells, causing alterations in genetic material and therefore stimulating the development of certain cancers [5,6,7]. Loss-of-function mutations lead to a decreased expression of tumor suppressor genes, resulting in the diminishment of tumor-protective functions [8,9,10,11]. The collective data obtained suggest that distinct pathways of genetic alteration lead to cancer [11]. Products of both oncogenes and tumor suppressor genes regulate cell cycle machinery [8,12]. There are different phases of the cell cycle, and progression through these phases requires many regulatory components, which include oncogenic genes (CDKs (cyclin-dependent kinases) and cyclins) and tumor suppressor genes (cyclin-dependent kinase inhibitors) [13]. The identification and subsequent functional analysis of cyclins and CDKs enable us to understand the molecular mechanism of cell-cycle machinery [14]. In the pathogenesis of breast cancer, cyclins A, B, C, D1, and E; CDKs; and CDK-inhibitors, such as p21 (Waf1/Cip1), p27 (Kip1), p16 and 53, are known to play important roles in cell cycle control [15] (Figure 1). Each cell cycle phase is tightly regulated in normal cells [15]. After exposure to mitogenic stimuli, however, these regulatory components become deregulated, which predisposes the cellular transformation of breast epithelial cells. Numerous studies implicate the roles of oncogenic and tumor-suppressive components in various human cancer types, including the initiation and development of breast cancer, more specifically [9,16,17]. In addition, several research bodies have confirmed the prognostic significance of oncogenic and tumor suppressor components in regard to therapy or clinical outcomes [18]. Significant information exists on the regulation and roles of the cell cycle components in breast cancer cells, and previous studies may be utilized for therapeutic purposes. Findings from experimental studies also support that alterations in these components are clinically significant [19]. The aim of this review article is to summarize the role of various oncogenic and tumor suppressor components of the cell cycle that are involved in breast cancer progression and prognosis.

## 2. Overview of Cell Cycle

The cell cycle is composed of several phases (Figure 1), including a phase for the preparation of DNA synthesis—G1 phase; a phase for DNA synthesis—S phase; a second preparation phase—G2 phase; and mitosis—M phase. These phases are tightly controlled under physiological conditions [20]. Quiescence (G0) is another phase of the cell cycle found in some differentiated cells, in which the cell undergoes its own distinct biochemical or molecular changes [21]. Under certain pathological stimuli, differentiated cells can leave the G0 phase and re-enter the cell cycle [21]. The transitions between these cell cycles phases are controlled by the function of specific CDKs. This includes CDK1/CDK2, which causes the transition from G2 to mitosis, and CDK2/CDK4/CDK6, which causes the G1 to S phase transition [22]. During cellular division, another group of proteins called cyclins form complexes with specific CDK molecules in their respective phases [22]. The G1 cell cycle phase transition is identified by the activity of CDK4/6-cyclin D and CDK2-cyclin E complexes, S cell cycle phase transition by cyclin A-CDK1/2 complex, and G2-mitosis phase transition by cyclin A-CDK and cyclin B-CDK1 complexes [23]. Many genetic alterations can affect the functional activities of oncogenes or tumor suppressors, including alterations in cyclin E, cyclin D1, and p27. These alterations have been shown to induce a transition from the quiescent state into the active state in breast epithelial cells, subsequently leading to breast epithelial cell transformation (Figure 2) [24].

## 3. Oncogenic Components of Cell Cycle

### 3.1. Cyclin D

The proto-oncogene cyclin D is a crucial regulator for the transition from the G1 to S phase during the cell cycle. It binds with CDK4 and CDK6 and forms an active cyclin D-CDK4/6 complex, which then phosphorylates the retinoblastoma protein (Rb) to promote cell cycle progression [25,26]. Cyclin D may also modulate the activity of various transcription factor proteins and histone deacetylase enzyme [27]. Having a half-life of ~24 min, cyclin D is degraded inside the cell mainly via the activity of 26S proteasome in a ubiquitin-dependent and Skp2 F-box protein-dependent manner [28,29]. In addition, D1-CDK4/6 complex can also impair the functions of mitochondria through the phosphorylation and repression of nuclear respiratory factor 1 (NRF1) and mitochondrial transcription factor A (mtTFA). An earlier report established a molecular link between cyclin D1 and control of mitochondrial function through the inhibition of nuclear respiratory factor 1 [30]. Previously accumulated data underscore the role of cyclin D1 in the tumorigenesis of mammary cancer [31,32]. Overexpression and gene amplification of cyclin D has also been linked to a worsened prognosis and the development of resistance against endocrine therapy in breast cancer (Table 1 and Table 2) [33,34]. A study documented cyclin D1 gene overexpression and copy number amplification in 20% and 50% of human breast cancer cases, respectively [35,36,37]. Furthermore, an enhanced expression of cyclin D1 was also observed in 67.5% of invasive ductal carcinoma cases, where it was strongly correlated with estrogen receptor (ER) and progesterone receptor (PR) expression [38]. Similarly, a study analyzed the immunohistochemical (IHC) positivity of cyclin D1 in invasive ductal and moderately differentiated breast cancer cases, which was associated with significantly poorer prognoses in these patients [39]. Additionally, research data based on in vitro and clinical studies implicated an increased cyclin D1 gene expression and amplification in ~45–50% of breast cancer cases [40]. In another in vitro study, genetic alterations in the cyclin D1 gene and mRNA expression were found in the ER-negative MDA-MB-453 cell line (Table 3), which may be related to malignant transformation [41]. Similarly, cyclin D1 protein expression was examined in infiltrating mammary carcinoma with ER/PR positivity [42]. An abnormal expression of cyclin D1 was displayed in 66% of mammary infiltrating duct carcinomas, suggesting its role in breast tumor metastasis [43]. Zhang et al. [44] determined that enhanced expression of the cyclin D gene was found in ~82% of human breast tumors, and gene amplification was present in ~17% of cases.

A study using a mouse mammary tumor virus model of breast cancer identified CCND1 gene amplification with positive IHC staining in 40% of breast cancer samples [132]. Further, the study identified ectopic overexpression of cyclin Dl and a reversed growth-inhibitory outcome after anti-hormonal therapy in ER-positive breast cancer cases, which provided a potential antitumor mechanism [46]. Kenny et al. [43] showed that ER-positive breast cancer patients had cyclin D1 high expression, and at the same time, also displayed more risk of relapse, metastasis, and early death [34]. Moreover, the data also showed that CCND1 gene amplification alone is a strong predictor of anti-hormonal therapy response in young-age breast cancer patients [47]. Moreover, data from another study indicated amplification of the cyclin D1 gene and noted its correlation with ER-positive invasive lobular breast carcinoma with lymph node metastasis, suggesting a sign of poorer prognosis [48]. An additional study suggested overexpression of cyclin D1 gene in the ER-positive MCF-7 breast tumor cell line, which was responsible for hyperproliferation undergrowth factor-deprived conditions [114]. Another study identified overexpression of cyclin D1 in ER-positive and ER-negative breast cancer samples; however, both shorter overall survival and relapse-free survival were associated only with the ER-negative subgroup [49].

Correlation of high cyclin D1-related elevation with Rb phosphorylation was also observed in >100 high-grade breast carcinomas [115]. Furthermore, a separate study also demonstrated a strong positive correlation between cyclin D gene amplification and higher expression in basal-like and ER-positive breast cancer subtypes, and suggested that cyclin D1 was an independent predictor for prognosis in ER-positive breast cancers [50]. The ABCSG Trial 05 and 06 documented an increased expression of cyclin D1, which was associated with the poorer clinical outcome and shorter overall survival of breast cancer patients [51]. A separate investigation determined the cyclin D1 positivity in proliferative disease without atypia, atypical ductal hyperplasia, low-grade ductal carcinoma in situ (DCIS), high-grade DCIS, and invasive carcinoma. The results showed that cyclin D1 was significantly higher in proliferative disease than normal breast epithelium, and even higher in DCIS than proliferative disease [52]. Additionally, another research group demonstrated an association between high cyclin D1 gene expression and high-grade tumor development, increased Ki-67 expression, and poorer survival in the ER-positive breast cancer group [53].

The majority of invasive lobular carcinomas showed cyclin D1 overexpression at the protein levels, suggesting its role in the progression of invasive lobular carcinoma [54]. Another study showed that ER-positive patients with moderate cyclin D1 expression had benefited from anti-hormonal therapy (tamoxifen), whereas those with high cyclin D1 expression had not benefited from tamoxifen, suggesting its role as a predictive marker for tamoxifen resistance [55]. Further results suggest that the silencing of cyclin D1 expression may reduce the development and progression of tamoxifen-resistant tumors [116]. Cisplatin drug targets cyclin D1, and treatment of ER-positive MCF-7 breast cancer cells with cisplatin increased cell death or growth arrest by decreasing the cyclin D in MCF-7 cells [117]. Using techniques fluorescent in situ hybridization (FISH) and IHC, researchers observed that CCND1 had increased amplification in high-grade infiltrating ductal carcinoma in comparison to low-grade infiltrate ductal carcinoma [56]. Cyclin D1 overexpression has been found to have a strong correlation with receptor status, suggesting that cyclin D1 expression could be a biomarker for good prognoses [57,58].

Additionally, expression of cyclin D2 was found to be very rare in breast cancer cases, in comparison to normal human mammary epithelial cells [47,59]; its role in cancer is yet to be elucidated [133]. Cyclin D3 has also been reported to be overexpressed in breast cancer samples, but there are limited research data on its relationship to disease outcomes [118,133,134]. Furthermore, experimental evidence has also shown elevated cyclin D1 protein levels and deposition of cyclin D3 in breast cancer samples [60]. Another study identified that 64 breast cancer cases out of 82 had cyclin D1 gene amplification, and 36 out of 86 cases had cyclin D3 gene amplification [62]. Expression of cyclin D1 was evaluated in different molecular breast cancer subtypes, and results showed a stronger intensity of positive cyclin D1 staining in the ER positive/PR positive subtype than in triple-negative breast cancer (TNBC) cases, and negative cyclin D1 staining was seen in human epidermal growth factor receptor 2-positive (HER2-positive) molecular subtypes. Further, TNBC cases with a low amount of cyclin D1 expression had higher tumor grade, tumor stage, and more positive lymph nodes with lymphovascular invasion, proposing that cyclin D1 expression may be a key factor to consider for aid in breast cancer management [61]. Lundberg et al. [135] determined CCND1 amplification and its association with worst 15-year survival with ER+/LN−/HER2−(1.66; 1.14–2.41), luminal A (HR = 1.68; 95% CI, 1.15–2.46), and luminal B (1.37; 1.01–1.86) breast cancer subtypes [135]. Overexpressed cyclin D1 induced Dicer expression in luminal A and basal-like breast cancer subtypes [136]. In another study, lower levels of cyclin D led to a decrease in MDA-MB-231 cells’ motility which resulted due to the decrease in phosphorylation of filamin A protein [137]. These studies found that cyclin D1 can also contribute to cellular proliferation and migration through non-canonical functions.

### 3.2. Cyclin A

Cyclin A protein forms complexes with both CDK1 and CDK2, which functions in both the S to G2 phase transition and the G2 to M phase transition of the cell cycle [51]. In the S phase, the cyclin-A-CDK complexes phosphorylate the components of the DNA replication machinery, subsequently initiating replication [51]. While in the mitosis phase, cyclin A/CDK2 coordinates centrosomal and nuclear mitotic events. However, it is thought to contribute to the stability of other cyclin molecules [51]. The increased expression of cyclin A gene has been found in different types of human tumors, including breast cancer, which suggests that cyclin A may potentially serve as a prognosis marker for the disease (Table 1 and Table 2). Studies have shown that microinjection of cyclin A into *Xenopus* oocytes and mammalian cells stimulates the breast tumor epithelial cells and induce the transition into M phase of the cell cycle [46,58,119,138,139]. A great number of tumors have shown a strong statistical correlation between cyclin A gene amplification and cyclin A protein levels [62]. Findings suggested that assessment of cyclin A and/or E2-promoter binding factor 1 (E2F1) expression levels associated with Ki-67 might be a useful tool for improved prognostic evaluation in negative lymph node breast cancer patients [63]. Another study showed that cyclin A is an independent prognostic factor and predictor of both breast cancer recurrence and response to tamoxifen therapy [64]. Lastly, overexpressed cyclin A was observed to be significantly correlated with breast cancer patients with earlier relapse, higher risk, and shorter overall survival rate, when compared to the breast cancer patients with better prognoses. Therefore, cyclin A may potentially be an accurate marker for tumor proliferation and prognosis in breast cancer [65].

### 3.3. Cyclin E

Cyclin E protein, a regulatory subunit for CDK-2, is thought to be a rate-limiting factor for the G1 to S phase cell cycle transition [140]. Cyclin E protein and its associated kinase (CDK2) experience well-regulated activation in normal cells. In actively-dividing tumor epithelial cells, however, the cyclin E and CDK complex remains activated throughout the cell cycle [141]. The deregulation in the expression of the cyclin E gene was found responsible for breast cancer tumorigenesis [46,60,114]. Previous data have demonstrated that higher levels of cyclin E gene amplification have been found in breast cancer tissues (Table 1) [62]. Another study observed an 8-fold amplification of the cyclin E gene and a 64-fold overexpression of its mRNA in human breast cancer cells, which provides evidence for aberrant cyclin E expression during tumorigenesis [120]. Further, a multivariate analysis correlated an elevated cyclin E level with poor patient outcome and showed that patients with elevated levels of cyclin E had a greater hazard ratio, as compared to those with low levels of cyclin E [142]. In addition, a relation between cyclin E gene expression and an ER-positive status was also observed in patients with breast cancer. In additional studies [89,90,91], cyclin E expression was greater in the ER-negative group and correlated with increased risks of death and relapse, suggesting that cyclin E may be responsible for ER-independent tumor growth. Similarly, cyclin E overexpression in breast cancer cells was associated with ER-negative tumors, HER2-positive tumors, and high-grade tumors with increased proliferation indexes [68,69]. A cohort study performed on 34 HER2-positive patients subjected to trastuzumab (Herceptin)-based therapy observed that the cyclin E gene copy number or mRNA overexpression was associated with diminished therapeutic benefits and lower rates of progression-free survival, as compared to non-overexpressing cyclin E patients [121]. Moreover, cyclin E expression was associated with a poor prognosis and closely related with cyclin D1 and p27Kip1 expression [70]. Similarly, high expression of cyclin E measured by IHC was a significant factor of poor prognosis and associated with a higher risk of death in the node-positive breast cancer group, as illustrated in a separate multivariate analysis [71].

### 3.4. Cyclin B

Two types of mammalian cyclin B regulate the G2-to-mitosis phase progression in the cell cycle, which do so by forming complexes with CDK1 kinase [143]. The available data suggest that breast cancer patients experience cyclin B gene amplification and overexpression at both the mRNA level and protein level (Table 1 and Table 2) [62]. Its increased expression has been correlated with a large tumor size, a high tumor grade, lymph node involvement, an ER-negative/PR-negative status, and a HER2-positive status [74]. Its overexpression has also been linked with younger age at diagnosis and higher expression levels of cyclin A, cyclin E, and Ki-67 [144]. Both univariate and multivariate analyses significantly identified an increased breast cancer death rate correlated with cyclin B1 overexpression, suggesting that it serves as a remarkable prognostic factor [72].

A meta-analysis investigated the significance between cyclin B protein and clinicopathological characteristics in breast cancer patients. Observations showed that overexpressed cyclin B was associated with poorer rates in disease-free survival (DFS), disease-specific survival (DSS), and overall survival (OS), along with a positive association with lymphatic invasion [73]. Androic et al. [100] observed apoptosis induction and growth reduction in different breast cancer cell lines, namely MCF-7, MDA-MB-231, BT-474, and SK-BR-3 (Table 3), in the absence of cyclin B. The suppression of cyclin B via small interfering RNA (siRNA) caused G2/M cell cycle phase arrest in breast cancer cell lines [73,125]. The HER2-positive invasive breast cancer samples used for the determination of cyclin B1 expression showed a direct correlation between positive cyclin B1 staining and higher tumor grade, large tumor size, positive lymph node counts, younger age, and higher Ki-67 expression. Thus, due to its relation with an aggressive phenotype, cyclin B1 might be considered a strong independent prognostic factor in breast cancer [74].

### 3.5. CDK2

Cyclin-dependent kinase 2 binds and forms complexes with cyclin E or cyclin A proteins and exclusively promotes the G1 to S and G2 to M phase transition within the cell cycle [145,146]. It has been observed that fulvestrant inhibited cyclin E-CDK2 activity, which in turn promoted the arrest of MCF-7 cells in quiescence (G_0_) [147]. Similarly, the findings suggest that the suppression of the cell-cycle progression through the G1 cell cycle phase by pentagalloylglucose (5GG) treatment in MCF-7 cells was mediated by blocking cyclin E/CDK2 activity [122].

## 4. Tumor Suppressive Components of Cell Cycle

### 4.1. p16(INK4A/MTS-1/CDKN2A)

The tumor suppressor p16, also known as INK4A/MTS-1/CDKN2A, has widespread importance in oncology due to its CDK-inhibitory function [148]. The frequently occurring SNP (single nucleotides polymorphism) mutations and deletions of the p16 gene in breast cancer cells suggest an important role in tumorigenesis [149]. The p16 protein molecule binds to and inactivates the cyclin D-CDK4/6 complexes, leads to subsequent Rb protein inactivation, and consequently results in cell cycle arrest [149]. Archived breast tumors of different histological subtypes provided evidence that aberrant p16 gene expression is the most common abnormality in human breast cancer (Table 1) [150]. Furthermore, an abnormal expression of p16 was found in ER-negative, pre-menopause breast cancer patients, in comparison to ER-positive patients. The abnormal p16 expression these researchers observed was closely associated with a high proliferative index [108]. An earlier study suggests that abnormal p16 expression may act as a predictor of poor response to hormonal therapy [92]. Another study found p16 protein-positive expression in the luminal A subtype of breast cancer patients, and higher expression was associated with breast cancer progression from DCIS to invasive ductal carcinoma (IDC) [96]. Abou-Bakr et al. [110] investigated the p16 expression in basal-like breast carcinoma grade III with histopathological findings in line with IDC. Results suggested that the p16 protein demonstrated high IHC intensity in basal-like carcinoma, which subsequently was associated with brain and lung metastasis [97]. A study by Arima et al. [126] found low p16 expression in resistant TNBC carcinoma [126]. Both p16-positive and p16-negative cells in the stromal cells of invasive lobular carcinoma reflected high nodal involvement, early recurrence, and metastatic propensity. Additionally, restoration of p16 expression in stromal fibroblasts suppressed cancer cell migration and invasion. Thus, these findings proposed positive stromal p16 expression as a treatment strategy to prevent nodal or distant metastasis [98].

### 4.2. p21 (WAF1/CIP1/SDI1/MDA-6)

The CDK-inhibitor p21 (also known as WAF1/CIP1/SDI1/MDA-6) activates the CDK4 and proliferating cell nuclear antigen, which results in G1 phase arrest [151]. Both in vivo and in vitro experimental models demonstrated that overexpression of p21WAF1/CIP1 resulted in G1 cell cycle phase arrest and effectively suppressed tumor growth (Table 1 and Table 2) [151]. Data on lymph node-negative breast cancer patients suggested that detection of p21 indicates the presence of a parameter that may act as a tumor suppressor and benefit patient survival [75]. Another study identified p21-positive tumor cell nuclei in more than 30% of the breast carcinomas, which was remarkably associated with a low histological grade and node-negative status [76]. The findings strongly suggested that p21WAF1/CIP1 gene expression might be used as a key prognostic biomarker for breast cancer, allowing therapy options to be adjusted more appropriately for individual cancer patients [77]. Breast cancer mastectomy used for measuring p21WAF1/CIP1 expression showed its upregulation in larger tumors in patients who presented with higher tumor dedifferentiation grades, more lymph node metastases, and shorter disease-free survival rates [78]. Moreover, an in vitro study where ER-positive or ER-negative breast cancer cell lines were immunostained for evaluation of p21 found a direct correlation between p21WAF1/CIP1 and ER expression [123,124].

In addition, p21WAF1/CIP1 also plays multifaceted roles in breast cancer. For instance, p21WAF1/CIP1 expression induced cell invasion and had correlation with OS and distant metastasis-free survival in breast cancer patients mediated via controlling TGFβ/Smad signaling [152]. A study measured high p21WAF1/CIP1 levels in the cytoplasm of metastatic breast cancer cells where it was associated with elevated p53 levels and poor prognoses [153]. Multiple studies identified that phosphorylation of p21WAF1/CIP1 by AKT1 disrupted its binding with proliferating cell nuclear antigen (PCNA) and induced its cytoplasmic accumulation. Accumulated p21WAF1/CIP1 regulates the ERBB2-mediated proliferation of breast cancer cells and breast carcinogenesis [154,155]. Further, downregulation of p21WAF1/CIP1 promoted EMT, enhanced the cell viability and migration potential in response to long non-coding RNA plasmacytoma variant translocation 1 (PVT1) in distinct MDA-MB-231, MDA-BA-468 breast cancer cell lines [156]. Similarly, another study using breast cancer mouse models has shown that invasion is accompanied by an upregulation of p21WAF1/CIP1, indicating its oncogenic role [157]. The overexpression of p21WAF1/CIP1 has also been found to be associated with a poor response to tamoxifen treatment in MCF-7 cells [158]. Similarly, Akt-dependent phosphorylated p21WAF1/CIP1 enhanced doxorubicin resistance in SUM159 TNBC cells [159]. Another study demonstrated that p21WAF1/CIP1 inhibited apoptosis in breast cancer. The overexpression of p21WAF1/CIP1 in breast cancer decreased cell sensitivity to infrared-induced apoptosis through inhibition of CDKs [160].

### 4.3. p27 (Kip1)

Tumor suppressor p27, an important regulator for the G1 to S transition in the cell cycle, is known to coordinate the activation of the cyclin E-CDK2 complex with the accumulation of cyclin D-CDK4, which initiates the exit of cells from the cell cycle in response to anti-mitogenic signals [161]. The downregulation of p27 gene expression is strongly correlated with higher tumor grade and phenotypes with lower tumor differentiation (Table 1) [79]. Reduced levels of p27 protein is also an indicator of poor clinical outcomes in a majority of lymph node-negative breast cancer patients [79]. 

Multiple sources of evidence suggest that p27 induced G1 cell cycle phase arrest, mediated by transforming growth factor-β (TGF-β), rapamycin, and cyclic adenosine monophosphate (cAMP) [48,60,114,127]. Previous studies also demonstrated that high expression levels of p27 in human breast cancer cells inversely correlated with the degree of malignancy in the human breast [127]. Moreover, a high expression of p27 was noticed in breast cancer patients, which was significantly correlated with an ER-positive status and inversely associated with shorter survival [80]. A univariate Kaplan–Meier analysis indicated that the decreased expression of p27 was significantly correlated with a worse clinical course [81]. A flow cytometry study using resistant breast tumor cells demonstrated a higher S-phase fraction and increased CDK2 activity in low p27-expressed cells, which was reversed after an exogenous addition of p27 [128]. 

Immunostaining of breast tumor indicated that downregulation of p27 correlated with HER2 gene overexpression in primary breast carcinomas, which may be significant in selecting patients for HER2-positive/neu antibody therapy in the future [82]. A separate study found that tamoxifen treatment caused MCF-7 cell cycle arrest due to an upregulation of p27 levels [129]. Another evaluation of p27 expression observed that it was a significant predictor for 5-year breast cancer survival, and that reduced p27 expression correlated with a high histologic grade, an advanced TNM stage (tumor size, lymph node status, metastatic status), and negative hormone receptor status [83,84]. A reduced expression of p27 was also observed in docetaxel-resistant breast cancer cells (MCF-7 and MDA-MB-231 cell lines) [130]. Another univariate analysis showed a remarkable relationship between low p27 expression and increased tumor grade, nuclear pleomorphism, and mitosis, along with decreased tubule formation in ER-negative and ductal/no special type tumor status [85]. 

High p27 expression independently predicted superior relapse-free survival and overall survival, and subsequently suggested its use as an independent predictor in hormonal therapy response [86]. An immunohistochemically retrospective investigation of 216 breast carcinomas found that p27-negative patients had a poorer prognosis than those in other categories, highlighting that the examination of p27 expression may identify breast carcinoma patients who would benefit from adjuvant therapy [87]. Further, in the lymph node-negative population, decreased p27 immunoreactivity was associated with higher tumor grade, more HER2-positive overexpression, greater lymph node positive populations, lower expression of thymidylate synthase, higher Ki-67 expression, and poorer disease-free survival [88]. In hormonal receptor- positive carcinoma, lower p27Kip1 was correlated with decreased overall survival [hazard ratio (HR) = 1.42; 95% confidence intervals (CI) = 1.05 to 1.94; disease-free survival HR = 1.27; and 95% CI = 0.99 to 1.63], as compared to carcinoma with higher p27Kip1 expression treated with adjuvant therapy (doxorubicin and cyclophosphamide) [90]. An inverse correlation was also observed between p27Kip1 expression and the degree of breast tumor malignancy [162]. Breast cancer patients in Taiwan were evaluated for the expression of p27Kip1, and both univariate and multivariate analyses showed that lower p27Kip1 expression correlated with OS in ER/PR positive tumors. Therefore, p27Kip1 may be considered an independent prognosis marker for breast cancer in Taiwan [91]. 

Another meta-analysis study showed a significant association between high p27 expression and OS, DFS, and RFS in lymph node-negative and lymph node-positive breast cancer patients [163]. In addition, Austrian Breast and Colorectal Cancer Study Group Trial 06 enrolled early-stage breast cancer patients with an ER/PR hormonal-positive status for evaluation of p27Kip1 expression and observed its impact on the clinicopathological features of women receiving adjuvant tamoxifen for 5 years. Observations confirmed that high p27Kip1 expression was significantly associated with longer disease-free survival (0.22; 95% CI, 0.11–0.42; *p* < 0.001) and overall survival (0.39; 95% CI, 0.21–0.72; *p* = 0.002) as compared to women with low p27 expression [164].

### 4.4. p53 (Wild Type)

Tumor suppressor p53 protein plays a key role in coordinating the response of cells to several stress conditions, including oncogenic activation, hypoxia, and DNA damage [165]. In response to mitogenic stress, p53 activates apoptosis in normal cells. This same activation of apoptosis by p53 has also been observed in anticancer therapy response. A mutated version of p53 protein that does not respond appropriately during oncogenic stress allows cell transformation, resulting in tumor initiation [165]. After immunohistochemical evaluation of p53 expression in primary breast cancer specimens, it was assessed that p53 overexpression was associated with an advanced-stage tumor, metastatic spread, and lower concentrations of progesterone receptors (Table 1) [89]. An increased cytoplasmic accumulation of p53 was observed in breast cancer patients as well. These patient samples demonstrated high proliferative activity with median Ki-67 fractions increased by up to 75%, along with a 74% increase in median S-phase fraction compared to the control group [93]. 

Utilizing invasive ductal carcinoma samples, Yang et al. (2013) [143] calculated DFS and its correlation with p53 expression. The Cox regression and multivariate analysis showed that p53 expression acted as a predictive factor of DFS [100]. Additionally, several studies also associated positive p53 expression with worsened prognoses. For instance, a Kaplan–Meier analysis of TNBC invasive ductal carcinoma samples showed that a positive p53 expression was correlated with worse overall survival (79.6% vs. 89.6%, log-rank test *p* = 0.025) and the patients had a 2.2 times higher mortality risk than that of p53-negative patients (HR: 2.222; 95% CI: 1.147–4.308) [102]. Similarly, p53 overexpression tested by IHC on modified radical mastectomy samples obtained from TNBC patients also showed lower overall survival rates (*p* = 0.021, log-rank test) compared to the patient group with low p53 expression. Moreover, the multivariate analysis proposed p53 overexpression as having the strongest prognostic significance in TNBC patients (<50 years) [103]. 

In a retrospective study of a large number of luminal/HER2-negative breast cancer patients, the data demonstrated that a p53 expression of ≥50% (present in 9% patients) was associated with shorter disease-free survival, in comparison to patients with p53 expression of <50%. Therefore, p53 overexpression was classified as a prognostic marker for unfavorable characteristics [104]. Another study performed on ER-positive and ER-negative invasive breast cancer determined an association of p53 overexpression with ER status. Results showed that in ER-negative breast tumors, a higher p53 expression was associated with DFS and OS than in ER-positive breast tumors [105]. 

Expression of the p53 gene was also evaluated in all breast cancer subtypes, i.e., luminal A, luminal B, HER2-positive, TNBC, and basal-like, and the findings suggested that p53 had a higher expression within HER2-positive and TNBC subtypes than in luminal A and luminal B subtypes. The overexpression of p53 in HER2-positive and TNBC subtypes also had significance in early-onset, high-grade tumors, and an increased proliferative index [106]. In invasive breast carcinoma grade II and III samples, positive p53 expression was significantly related with increased tumor grade (*p* < 0.006), lymphovascular invasion (*p* < 0.003), and lymphocytic infiltration (*p* < 0.004). These results indicate that p53 overexpression is a marker for a poor prognosis and a compromised immune response in more aggressive breast cancer types [107]. 

Better overall survival was observed in p53-overexpressed TNBC cases than in p53-negative TNBC patients who underwent neoadjuvant chemotherapy [108]. Another study concluded that p53 overexpression was inversely correlated with ER/PR expression and positively correlated with HER2-positive overexpression in high-grade tumors with nodal metastasis [109]. In a randomized stage II clinical trial on lymph node-positive patients who received four cycles of cyclophosphamide and one dose of doxorubicin adjuvant therapy, epithelial p53 expression was evaluated (using monoclonal antibodies DO7 and 1801). After univariate analysis, this study stated that positive p53 IHC was associated with worse OS and RFS in lymph node-positive patients [110].

### 4.5. p53 (Mutant)

A study performed by Marchetti et al. [154] found an “Arg72Pro” p53 variant in 23% of primary breast cancer patients. The patients positive for the Arg72Pro variant had relapsed within 10 months of the median DFS, compared to those that showed a wild-type p53 status [111]. Lenora W.M. et al. [155] also found a higher nuclear expression of mutant p53 using PAb1801 monoclonal antibody in young breast cancer patients. Kaplan–Meier curves and a log-rank test analysis correlated mutant p53 expression with a poor prognosis among distinct ethnic populations. Similarly, TNBC patients with abnormal mRNA expression of mutant p53 in a separate study were more likely to experience less 5-year reoccurrence-free survival. Mutant p53, therefore, may be considered a potential prognostic marker in TNBC patients [113].

## 5. Future Perspectives

Aside from great improvements in diagnostic tools and the increased availability of multiple therapeutic options, breast cancer cure rates remain poor. In the GLOBOCAN-2018 report, 2.1 million new breast cancer cases (11% of all total cancer types) were diagnosed in 185 countries [166]. In India, 144,000 breast cancer cases with a 5-year prevalence of 396,000 and 70,000 deaths were reported in 2012 [166,167]. As per the GLOBOCAN-2018 report, 162,000 breast cancer cases (27.7% of all new cancers), a 5-year prevalence of 405,000 breast cancer cases, and 87,000 deaths were observed in the Indian population [166]. Knowledge of reliable biomarkers related to disease prognosis and therapy decisions can improve cancer management. In this regard, the oncogenic and tumor suppressor components of the cell cycle may serve as such markers. Data from the multiple studies provided above support this notation.

Although these markers are detectable by expression-profiling experiments, the lack of reproductivity of the described results from various studies delays their use in the clinical setting. In addition to technique sensitivity issues, the non-reproducibility of the results might be due to the variation in sample selection methods or variations in study designs. Studies having a smaller number of patients, different-aged patients, varying tumor grades, varying tumor sizes or metastatic potential, and different patient ethnicities can also lead to non-reproducible results. Therefore, larger-scale validation studies involving greater demographic, ethnic, and clinicopathological variabilities are required in the future, before we can apply their suggestive use in patient management. High throughput technologies, such as next-generation RNA sequencing [168,169,170] and single-cell RNA sequencing [171,172], give results with high coverage and depth, and cover potential sensitivity issues. The use of these technologies may help to identify reliable biomarkers for cancer management. Thus, the data presented in the review article propose the use of cell cycle components as biomarkers in breast cancer management.

## 6. Conclusions

Previous experimental studies have described several oncogenic and tumor-suppressive genes involved in cell cycle regulation and progression among various subtypes of human breast cancer. It is well-established that multiple genetic alterations are required for tumorigenesis, yet continued research regarding the specific and sequential mechanisms involved—and how they affect clinical outcomes—may continue to guide new therapeutic strategies for more effective cancer treatments. Current research supports the notion that these regulatory cell cycle genes are useful prognostic biomarkers in breast cancer tumorigenesis, and the clinical relevance of these suggestive biomarkers has been established by several studies, as described above. The accurate measurement of cell cycle component expression and their correlation with clinical symptoms and prognoses may provide valuable insight for the future of both breast cancer management and anti-cancer therapeutics.

## Figures and Tables

**Figure 1 pharmaceutics-13-00569-f001:**
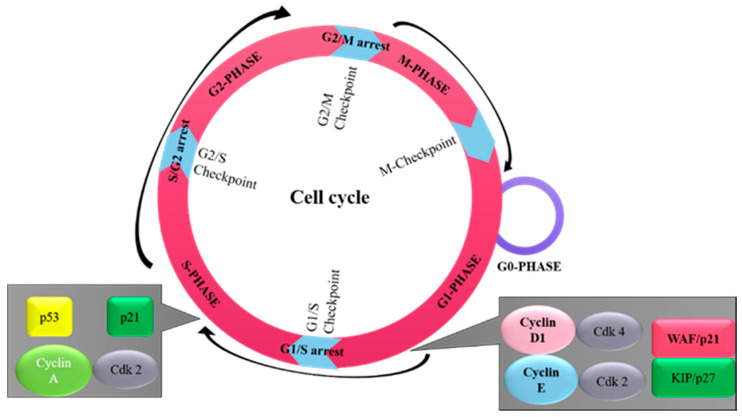
The sequential order of cell cycle events. The cell cycle progresses through four sequential phases: G1-phase (cell increases in size), S-phase (DNA synthesis), G2-phase (prepares to divide), and M-phase (cell division). The phases G1, S, and G2 make up the interphase stage, and span between cell division. There are special proteins and checkpoint systems for the proper progression of the cell cycle. First: G1 checkpoint (at G1/S transition) is the main irreversible decision point for cell division, which assesses for adequate cell size, availability of nutrients, positive molecular signals, and DNA integrity. Second: G2 checkpoint (at G2/M transition) ensures smooth cell division and assesses DNA integrity and successful DNA replication before division. In the case of error, cellular progression will become paused at the G2 checkpoint for repair. Third: the spindle checkpoint (metaphase to anaphase transition), ensures correct attachment of sister chromatids to the spindle microtubules.

**Figure 2 pharmaceutics-13-00569-f002:**
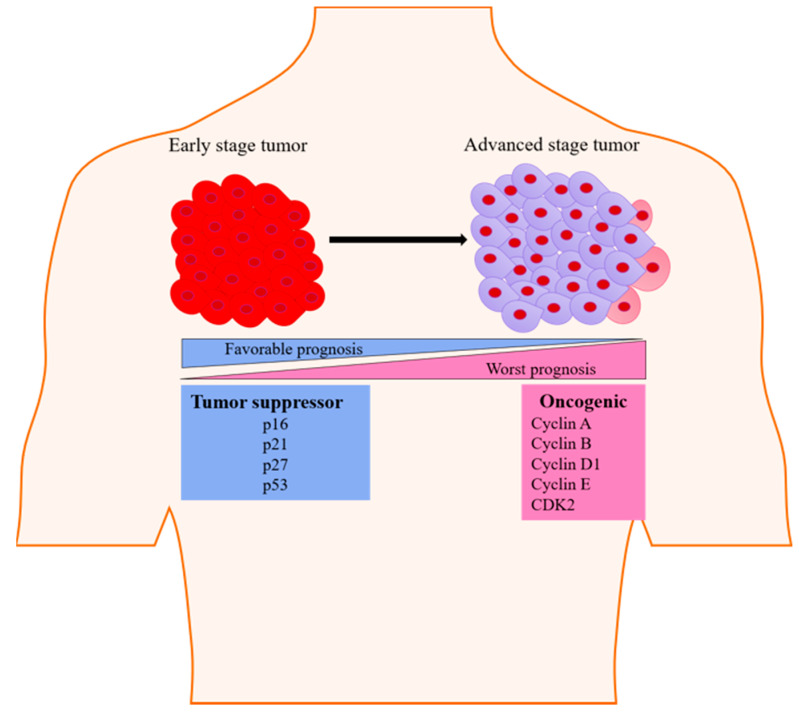
Illustration showing the change in the expression status of genes serving as tumor-suppressive and oncogenic markers during tumorigenesis. These changes in expression impact progression into advanced-staged cancer and overall breast cancer prognosis.

**Table 1 pharmaceutics-13-00569-t001:** Clinical relevance of oncogenic and tumor suppressive cell cycle components in breast cancer patients with different molecular subtypes.

Marker	Expression	Consequences	Receptor Status	Ref
Cyclin D	Overexpression	High risk of replace, Local reoccurrence, Metastasis	ER+/ER-	[45]
	Overexpression	High tumor grade	ER+/ER-/PR+/PR-/HER2+	[35]
	Overexpression	High proliferation	ER+/ER-/PR+/PR-/TNBC	[38]
	Overexpression	High proliferation	ER+/ER-/PR+/PR-	[39]
	Overexpression	High proliferation	ER+/ER-	[40]
	Overexpression	High proliferation	ER+/ER-/PR+/PR-	[42]
	Overexpression	Metastasis		[43]
	Gene amplification Overexpression	High proliferation		[44]
	Overexpression	High proliferation	ER+	[46]
	Gene amplification	High risk for recurrence	ER+/ER-	[47]
	Overexpression, Gene amplification	High proliferation	ER+/ER-/PR+/PR-	[48]
	Overexpression	High proliferation, Short overall survival, Large tumor size, Lymph node metastasis	ER+/ER-/PR+/PR-	[49]
	Overexpression, Gene amplification	High proliferation	ER+/ER-/Basal like	[50]
	Overexpression	Reduce relapse-free survival	ER+/PR+	[51]
	Overexpression	High proliferation	ER+/PR+/ER-/PR-/HER2+	[52]
	Overexpression	High proliferation	ER+/PR+/ER-/PR-/HER2+	[53]
	Overexpression	High proliferation	ER+/ER-	[54]
	Overexpression	High risk of recurrence	ER+ER-/HER2+/HER2-	[55]
	Gene amplification	High proliferation	ER+/ER-	[56]
	Gene amplification	Reduce patient survival time, therapy resistance	ER+	[57]
	Overexpression	Poor prognosis	ER+/PR+/ER-/PR-/HER2+/Basal like	[58]
	Overexpression	Invasiveness, metastasis	TNBC	[59]
	Overexpression	High proliferation	ER+	[60]
	Reduce expression	High tumor grade, Nodal positive status, Invasion	ER+/PR+/ER-/PR-/HER2+/HER2-	[61]
Cyclin A	Overexpression, Gene amplification	Poor prognosis		[62]
	Overexpression	Relapse, Shorter disease-free survival	ER+/ER-	[63]
	Overexpression	Worst prognosis	ER+/ER-	[64]
	Overexpression	Shorter relapse time	ER+/ER-	[65]
	Overexpression	Less survival rate, High relapse rate	ER+/PR+/ER-/PR-	[66]
	Overexpression	Short distant metastasis-free survival	ER+/PR+/ER-/PR-	[67]
	Overexpression	Poor prognosis	ER+/ER-	[68]
	Overexpression	High tumor grade, High proliferation index	HER2+/HER2-	[69]
	Overexpression	Poor survival	ER+/ER-	[70]
	Overexpression	Poor prognosis, Decrease survival rate	ER+/PR+/ER-/PR-/HER2+	[71]
Cyclin B1	Overexpression	Decrease survival	ER+/PR+/ER-/PR-/HER2+/HER2-	[72]
	Overexpression	Reduce overall survival, Disease free survival, Lymphatic invasion	ER+/PR+/ER-/PR-/HER2+/HER2-	[73]
p21(WAF1/Cip1)	Overexpression	High tumor grade, Large tumor size, Positive lymph node, High Ki-67 expression	ER+/PR+/ER-/PR-/HER2+/HER2-	[74]
	Overexpression	Favorable prognosis	ER+/PR+/ER-/PR-	[75]
	Overexpression	Better survival	ER+/PR+/ER-/PR-/HER2+	[76]
	Overexpression	Better survival	ER+/PR+/ER-/PR-/HER2+/HER2-	[77]
	Overexpression	Large tumor size, High tumor grade, Lymph node metastasis	ER+/PR+/ER-/PR-/HER2+/HER2-	[78]
p27 (Kip1)	Reduced expression	High tumor grade, Lack of tumor differentiation, Poor prognosis	ER+/PR+/ER-/PR-/HER2+/HER2-	[79]
	Overexpression	Better prognosis	ER+/ER-	[80]
	Overexpression	Favorable prognosis	ER+/PR+/HER2+	[81]
	Reduced expression	Poor prognosis	ER+/PR+/ER-/PR-/HER2+/HER2-	[82]
	Overexpression	Long disease-free survival, overall survival	ER+/PR+/ER-/PR-	[83]
	Reduced expression	Poor prognosis	ER+/PR+/ER-/PR-	[84]
	Reduced expression	Large tumor size, high tumor grade, lymph node metastasis	ER+/ER-	[85]
	Overexpression	Long relapse-free survival, Overall survival	ER+/PR+	[86]
	Reduced expression	Poor prognosis	ER+/PR+	[87]
	Reduced expression	Increase proliferation	ER+/PR+/ER-/PR-/HER2+/HER2-	[88]
	Overexpression	Favorable prognosis	ER+/PR+/ER-/PR-/HER2+/HER2-	[89]
	Lower expression	Worst overall survival, Worst disease-free survival	ER+/PR+/ER-/PR-	[90]
	Lower expression	Worst overall survival	ER+/PR+/ER-/PR-	[91]
p16 (ink4a)	Overexpression	High proliferation index	ER+/PR+/ER-/PR-/HER2+/HER2-	[92]
	Overexpression	Favorable prognosis	ER+/PR+/ER-/PR-	[93]
	Overexpression	Favorable prognosis	ER-/PR-/HER2-	[94]
	Overexpression	Favorable prognosis	ER+/PR+/ER-/PR-	[95]
	Overexpression	Disease progression	ER+/PR+/ER-/PR-	[96]
	Overexpression	Lung and brain metastasis	ER+/PR+/ER-/PR-/HER2+/HER2-	[97]
	Reduced expression	Metastasis	ER+/PR+/ER-/PR-/HER2+/HER2-	[98]
p53 (wild type)	Overexpression	Favorable prognosis	ER+/PR+/ER-/PR-/HER2+/HER2-	[99]
	Overexpression	Better disease-free survival	ER+/PR+/ER-/PR-	[100]
	Overexpression	Favorable prognosis	ER+/PR+/ER-/PR-/HER2+/HER2-/Basal like	[101]
	Overexpression	Favorable prognosis	ER+/PR+/ER-/PR-	[102]
	Overexpression	Worst prognosis	ER-/PR-/HER2-	[103]
	Overexpression	Worst prognosis	ER+/PR+/ER-/PR-/HER2+/HER2-	[104]
	Overexpression	Worst prognosis associated with ER expression	ER+/PR+/ER-/PR-	[105]
	Overexpression	Worst prognosis associated with HER2+/TNBC subtypes	ER+/PR+/ER-/PR-/HER2+/HER2-	[106]
	Overexpression	Worst prognosis, High tumor grade, Lymph vascular invasion, Lymphocyte infiltration	ER+/PR+/ER-/PR-	[107]
	Overexpression	Better overall survival in TNBC	ER-/PR-/HER2-	[108]
	Overexpression	Correlated with HER2 overexpression, High tumor grade	ER+/PR+/ER-/PR-/HER2+/HER2-	[109]
	Overexpression	Worst overall survival, Reoccurrence free survival	ER+/PR+/ER-/PR-	[110]
p53 (Mutant)	Overexpression	Early relapse	ER+/PR+/ER-/PR-/HER2+/HER2-	[111]
	Overexpression	Poor prognosis	ER+/PR+/ER-/PR-	[112]
	Overexpression	Less 5-years reoccurrence free survival	ER+/PR+/ER-/PR-	[113]

ER: estrogen receptor; PR: progesterone receptor; Her2+: human epidermal growth factor receptor-2 positive.

**Table 2 pharmaceutics-13-00569-t002:** Clinical relevance of oncogenic and tumor suppressive cell cycle components in breast cancer cell lines with different molecular subtypes.

Marker	Expression	Consequences	Model	Histology	Ref
Cyclin D	Gene amplification mRNA Overexpression	Proliferation	MCF-7	ER+	[33]
	Overexpression	Malignant transformation	MDA-MB-453	HER2+	[41]
	Overexpression	Increase proliferation	T-47D, MCF-7	ER+/HER2+	[34]
	Overexpression	Increase proliferation	MCF-7	ER+	[114]
	Overexpression	Increase proliferation	MCF-7, T-47D, MDA-MB-468, BT-549	ER+/ER-/PR-/HER2-	[115]
	Overexpression	Increase proliferation	HBL-100, MDA-MB-23l, T-47D, MCF-7, MDA-MB-134, HMEC-184	ER+/PR+/ER-/PR-/HER2-/HER2+	[52]
	Overexpression	Increase proliferation	MCF-7T,	Tamoxifen-resistant	[116]
	Downregulation	Cell death, Growth arrest	MCF-7, MDA-MB-231, MDA-MB-435, HCC-1937, CAL-148	ER+/ER-/PR-/HER2-	[117]
	Overexpression	Invasiveness, metastasis	MDA-MB-231	ER-/PR-/HER2-	[59]
	Overexpression	Increase proliferation	MCF-7	ER+	[118]
	Overexpression	Increase proliferation	MCF-7	ER+	[60]
	Overexpression	Increase proliferation	MCF-7	ER+	[119]
	Overexpression	Increase proliferation	ZR75-1-2, ZR-75-1, MDAMB-157, MDA-MB-231, MDA-MB-436, T-47D, BT-20, HBL-100, Hs578T, SK-BR3	ER+/ER-/PR+/PR-/HER2-/HER2+	[120]
Cyclin E	Overexpression, Gene amplification	Increase proliferation	BT-474, BT-474R	HER2+	[121]
	Downregulation	Suppression of cell cycle progression	MCF-7	ER+	[122]
Cyclin B	Overexpression	Associated with ER+ status	MCF-7, MDA-MB-231, MDA-MB-436, Hs578 T	ER+/PR+/ER-/PR-/HER2-/HER2+	[123,124]
	Downregulation	Apoptosis, Anti-proliferation	MCF-7, BT-474, SK-BR-3, MDA-MB-231	ER+/ER-/PR+/PR-/HER2-/HER2+	[125]
p16(WAF1/Cip1)	Reduced expression	Therapy resistance	HCC-1428, T-47D, MCF-7, MDA-MB-436, BT-549, MDA-MB-157, MDA-MB-231, MDA-MB-435S, Hs578T, HCC-1937, BT-20, SK-BR-3	ER+/PR+/ER-/PR^_^/HER2-/HER2+	[126]
	Overexpression	Good survival	MCF-7, BT-549, MDA-MB-134, MDA-MB-157, MDA-MB-231, MDA-MB-453, MDA-MB-468, ZR-75-1, BT-20, SK-BR-3, T-47D	ER+/PR+/ER-/PR-/HER2-/HER2+	[76]
p21(WAF1/Cip1)	Overexpression	High proliferation rate	ZR75-1, ZR75-30, MCF-7, MDA-MB-453, T-47D, Cal51, SK-BR-5, SK-BR-7, CAMA-1, BT-20	ER+/PR+/ER-/PR-/HER2+	[127]
	Reduced expression	Trastuzumab resistance	SK-BR-3	HER2+	[128]
	Overexpression	Cell cycle arrest	MCF-7	ER+	[129]
	Reduced expression	Acquired resistance to docetaxel	MCF-7, MDA-MB-231	ER+/ER-/PR-/HER2-	[130]
	Overexpression	Associated with ER+ status	MCF-7, MDA-MB-231, MDA-MB-436, Hs578T	ER+/PR+/ER-/PR-/HER2-/HER2+	[123,124]

**Table 3 pharmaceutics-13-00569-t003:** Characteristics of breast cancer cell lines with different molecular subtypes (adopted from Dai, X et al. 2017) [131].

Cell Line	ER	PR	HER2/neu+	Subtype	BRAC1 Mutation	p53 Mutation	Tumor
CAMA-1	+	+/-	-	LA	WT	MU	AC
HCC1428	+	+	-	LA	ND	ND	AC
MCF-7	+	-	-	LA	ND	WT	IDC
MDA-MB-134	+	-	-	LA	ND	MU	IDC
T-47D	+	+	-	LA	WT	MU	IDC
ZR75-1	+	+/-	-	LA	WT	WT	IDC
BT-474	+	+	+	LB	WT	MU	IDC
ZR75-30	+	-	+	LB	WT	WT	IDC
MDA-MB-453	-	-	+	Her2+	WT	MU	AC
SK-BR-3	-	-	+	Her2+	WT	MU	AC
SK-BR-5	-	-	+	Her2+	WT	MU	AC
BT-20	-	-	-	TNBC	WT	MU	IDC
BT-549	-	-	-	TNBC	WT	MU	IDC
CAL-51	-	-	-	TNBC	WT	MU	AC
CAL-148	-	-	-	TNBC	WT	MU	AC
HCC1937	-	-	-	TNBC	MU	MU	DC
Hs578T	-	-	-	TNBC	WT	MU	IDC
MDA-MB-157	-	-	-	TNBC	WT	MU	MC
MDA-MB-231	-	-	-	TNBC	WT	MU	AC
MDA-MB-435	-	-	-	TNBC	WT	MU	AC
MDA-MB-436	-	-	-	TNBC	MU	MU	AC
MDA-MB-468	-	-	-	TNBC	WT	MU	AC
SK-BR-7	-	-	-	TNBC	WT	WT	AC

WT: wild type; ND: not decided; MU: BRCA1 mutation; AC: adenocarcinoma; DC: ductal carcinoma; IDC: invasive ductal carcinoma; LA: luminal A; LB: luminal B; HER2+: human epidermal growth factor receptor 2-positive; TNBC: triple-negative breast cancer.

## Data Availability

Not applicable.

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
