# Peer review of "Oncogenic and Tumor Suppressive Components of the Cell Cycle in Breast Cancer Progression and Prognosis"

_pharmaceutics, 2021, doi:10.3390/pharmaceutics13040569_

Round 1

Reviewer 1 Report

Kashyap et al. present a review describing oncogenic and tumor suppressive components of the cell cycle with regards to breast cancer progression and prognosis. The review covers a broad number of studies but is descriptive and has many typographical errors. There are a number of issues that need to be addressed to improve the quality of the manuscript:

Major:

  1. There are excessive self-citations that are not essential/relevant to the topic of the review (phytochemical studies).
  2. Many words throughout the manuscript lack a space between them.
  3. Line 42, the phrase “These mitogenic stimuli make genetic changes” needs to be rephrased, as it is misleading.
  4. Lines 86-87, the specific CDKs for cyclin A should be mentioned.
  5. Table 1 is not very useful in its current format and should be condensed.
  6. Line 309, CDK2 was listed under section 3, oncogenic components, but this sentence describes a tumor suppressive function. Rather than just listing it, the authors need to explain these observations. This also applies throughout the manuscript where associations were described alongside anti-associations, but with no further narrative or explanation as to why that is the case.
  7. Line 323, p16 renders Rb inactive?
  8. Lines 496-500, the term lakh is not appropriate and should be replaced.
  9. Non-canonical roles of cell cycle components that are not related to the cell cycle should also be described, since these could also affect the associations described.

Minor:

  • Line 205 – “infiltering” needs to be corrected

Author Response

The authors of this manuscript express their sincere thanks to the reviewer for the critical assessment of this work. The authors have acted upon the recommendations of the reviewer which has resulted in a significant enhancement in the quality of this manuscript. All modifications incorporated in the manuscript are highlighted in red color font. A “point-by-point” response to each and every comment is outlined below.

General comments:

Kashyap et al. present a review describing oncogenic and tumor suppressive components of the cell cycle with regards to breast cancer progression and prognosis. The review covers a broad number of studies but is descriptive and has many typographical errors. There are a number of issues that need to be addressed to improve the quality of the manuscript:

Response:

We thank the reviewer for his/her expertise, time, and effort for reviewing our manuscript. We are deeply encouraged by the constructive comments and suggestions. As described below, we have revised our manuscript based on the reviewer’s worthy comments and recommendations.

Major:

Comment 1:

There are excessive self-citations that are not essential/relevant to the topic of the review (phytochemical studies).

Response:

We agree with the reviewer’s comments. Therefore, all the irrelevant citations are removed. 

Comment 2:

Many words throughout the manuscript lack a space between them.

Response:

We corrected all the typographical errors throughout the manuscript.

Comment 3:

Line 42, the phrase “These mitogenic stimuli make genetic changes” needs to be rephrased, as it is misleading.

Response:

Thank you for highlighting this sentence which has been rephrased (page number 2, line number 42 and 43).

Comment 4:

Lines 86-87, the specific CDKs for cyclin A should be mentioned.

Response:

Thank you for making this comment. We have now added the specific CDKs for cyclin A (page number 3, line number 86).

Comment 5:

Table 1 is not very useful in its current format and should be condensed.

Response:

We agree with the reviewer’s comment. Accordingly, this table has been condensed (page number 5). 

Comment 6:

Line 309, CDK2 was listed under section 3, oncogenic components, but this sentence describes a tumor suppressive function. Rather than just listing it, the authors need to explain these observations. This also applies throughout the manuscript where associations were described alongside anti-associations, but with no further narrative or explanation as to why that is the case.

Response:

We agree with the reviewer’s comment; therefore, this controversial line has removed from the manuscript (page 16).

Comment 7:

Line 323, p16 renders Rb inactive?

Response:

Thank you for this comment. We have rephrased this sentence (page 16, line numbers 331-333).

Comment 8:

Lines 496-500, the term lakh is not appropriate and should be replaced.

Response:

The unit lakh has been changed to appropriate number (page number 21, line numbers 526-530).

Comment 9:

Non-canonical roles of cell cycle components that are not related to the cell cycle should also be described, since these could also affect the associations described.

Response:

We agree with the reviewer’s comment. Therefore, non-canonical functions are added in the manuscript (page number 15, line numbers 226-233; page 17, line numbers 372-391).

Minor:

Comment:

Line 205 – “infiltering” needs to be corrected

Response:

“infiltering” has now been changed to “infiltrating” (page number 13, line number 207).

Additionally,

  1. The reference list has been modified as we have added several new references. Special attention is given to conform to the order of references and bibliographic style of the journal.
  2. The entire manuscript has been thoroughly checked and edited to ensure uniform style, organization, and quality.

On behalf of my co-authors, I once again express my sincere thanks to the erudite reviewer for the valuable suggestions and constructive input to improve the quality of our manuscript.

Reviewer 2 Report

Kashyap et al. tried to summarize the oncogenic and tumor suppressive components of “the” cell cycle in breast cancer progression and prognosis, which is a very fascinating topic, but it is also very challenging because it is a very broad topic.

Unfortunately, this reviewer would recommend to reject the review in the following form. It is a one-sided written review and concerning the p21 part it is incomplete, lacks important citations and the oncogene part of p21 is totally ignored. The localization of p21 is also very important. The reviewer would recommend the authors to read the papers of the Special Issue of Cancers (https://www.mdpi.com/journal/cancers/special_issues/p21) and the Reviews by Abbas and Dutta, Nat Rev Cancer, 2009 or Kreis et al. Oncogene, 2014. For the aspect “p21 and breast cancer”, this reviewer misses important references (Dai, M et al Breast Cancer Res. 2012; Vincent, A.J., FEBS Lett, 2012; Winters et al. 2001; to name a few), where the overexpression of “the tumor suppressor” p21 is not favorable for the prognosis and associated with chemoresistance. The illustration in Fig. 2. Should be reconsidered.

Line 345: Depending on the cellular context, p21 is able to activate OR inhibit Cdk4! p21 is also able to bind other Cdk family members, and it has also an important role during mitosis.

p21 has important functions in apoptosis and senescence, which should be mentioned.

Additional comments:

For table 3, the p53 status of the cell lines should be mentioned.

Line 231: The role of cyclin A in mitosis is NOT obscure. For example, Cyclin A/Cdk2 coordinates centrosomal and nuclear mitotic events, it is important for the timing of mitotic entry.

Line 307: Cdk2/cyclin A is also important for G2/M.

There are many missing spaces between words throughout the text. Therefore, it is sometimes difficult to read.

To conclude, the authors should probably focus on one target, like cyclin D. Here are also some publications missing like Lundberg, BCR, 2019.

Author Response

The authors of this manuscript express their sincere thanks to the reviewer for the critical assessment of this work. The authors have acted upon the recommendations of the reviewer which has resulted in a significant enhancement in the quality of this manuscript. All modifications incorporated in the manuscript are highlighted in red color font. A “point-by-point” response to each and every comment is outlined below.

Major comments:

Comment 1:

Kashyap et al. tried to summarize the oncogenic and tumor suppressive components of “the” cell cycle in breast cancer progression and prognosis, which is a very fascinating topic, but it is also very challenging because it is a very broad topic.

Response:

We thank the reviewer for his/her expertise, time, and effort for reviewing our manuscript. We are deeply encouraged by the constructive comments and suggestions. As described below, we have revised our manuscript based on the reviewer’s specific comments and recommendations.

Comment 2:

Unfortunately, this reviewer would recommend to reject the review in the following form. It is a one-sided written review and concerning the p21 part it is incomplete, lacks important citations and the oncogene part of p21 is totally ignored. The localization of p21 is also very important. The reviewer would recommend the authors to read the papers of the Special Issue of Cancers (https://www.mdpi.com/journal/cancers/special_issues/p21) and the Reviews by Abbas and Dutta, Nat Rev Cancer, 2009 or Kreis et al. Oncogene, 2014. For the aspect “p21 and breast cancer”, this reviewer misses important references (Dai, M et al Breast Cancer Res. 2012; Vincent, A.J., FEBS Lett, 2012; Winters et al. 2001; to name a few), where the overexpression of “the tumor suppressor” p21 is not favorable for the prognosis and associated with chemoresistance. The illustration in Fig. 2. Should be reconsidered.

Response:

We agree with the reviewer’s comments. Therefore, multifaced roles of p21 has been added to the manuscript (page 17, line numbers 372-391).

Comment 3:

Line 345: Depending on the cellular context, p21 is able to activate OR inhibit Cdk4! p21 is also able to bind other Cdk family members, and it has also an important role during mitosis. p21 has important functions in apoptosis and senescence, which should be mentioned.

Response:

We agree with the reviewer’s comment. Accordingly, apoptosis influencing role of p21 has been added (page 18, line numbers 389-391).

Additional comments:

Comment 1:

For table 3, the p53 status of the cell lines should be mentioned.

Response:

An additional column with p53 mutation status for each cell lines has been added to the Table 3 (page number 12)

Comment 2:

Line 231: The role of cyclin A in mitosis is NOT obscure. For example, Cyclin A/Cdk2 coordinates centrosomal and nuclear mitotic events, it is important for the timing of mitotic entry.

Response: 

We agree with the reviewer’s comment. The sentence has been rephrased (page number 14, line numbers 239 and 240). 

Comment 3:

Line 307: Cdk2/cyclin A is also important for G2/M.

Response:

The suggested information has been added (page number 16, line number 319).

Comment 4:

There are many missing spaces between words throughout the text. Therefore, it is sometimes difficult to read.

Response:

We have corrected all the typographical errors throughout the manuscript.

Comment 5:

To conclude, the authors should probably focus on one target, like cyclin D. Here are also some publications missing like Lundberg, BCR, 2019.

Response:

We agree with the reviewer’s comment. Therefore the, cyclin D section is further expanded (page 15, line numbers 226-233). The suggested reference has also been added (page 33, line numbers 980-1005).

Additionally,

  1. The reference list has been modified as we have added several new references. Special attention is given to conform to the order of references and bibliographic style of the journal.
  2. The entire manuscript has been thoroughly checked and edited to ensure uniform style, organization, and quality.

On behalf of my co-authors, I once again express my sincere thanks to the erudite reviewer for the valuable suggestions and constructive input to improve the quality of our manuscript.

Reviewer 3 Report

Minor revisions:

  1. Some words are merged together, to list a few: lines: 16, 25, 28, 38, 39, 45, 47, 48, 70, 184, 185, 210, 218, 236, 347. Please spell check and correct.
  2. Some grammatical errors are present, to name a few : line 70 there is “to summarizes”, should be “to summarize”; line 184 there is “which was it was”, should be “which was”; line 236 the is “induced”, should be “induces”; line 262 there is “cycle E”, should be “cyclin E”; line 275 there is “the expression”, should be “expression”; line 345 there is “in activates”, should be “activates”. Please correct.
  3. The first sentence of the introduction is copied from the abstract. Please re-write.
  4. Lines 210-211 “yet”; and line 371 “mediate” repeats twice. Please re-write.

Author Response

The authors of this manuscript express their sincere thanks to the reviewer for the critical assessment of this work. The authors have acted upon the recommendations of the reviewer which has resulted in a significant enhancement in the quality of this manuscript. All modifications incorporated in the manuscript are highlighted in red color font. A “point-by-point” response to each and every comment is outlined below.

Minor revisions:

 Comment 1:

Some words are merged together, to list a few: lines: 16, 25, 28, 38, 39, 45, 47, 48, 70, 184, 185, 210, 218, 236, 347. Please spell check and correct.

Response:

We corrected all the typographical errors throughout the manuscript.

Comment 2:

Some grammatical errors are present, to name a few: line 70 there is “to summarizes”, should be “to summarize”; line 184 there is “which was it was”, should be “which was”; line 236 the is “induced”, should be “induces”; line 262 there is “cycle E”, should be “cyclin E”; line 275 there is “the expression”, should be “expression”; line 345 there is “in activates”, should be “activates”. Please correct.

Response:

All the suggested corrections are made in the manuscript (page number 2, line number 70; page number 13, line number 186; page number 14, line number 246; page number 16, line numbers 272 and 285; page number 17, line number 355).

Comment 3:

The first sentence of the introduction is copied from the abstract. Please re-write.

Response:

The concerned sentence has been modified (page number 2, line numbers 38 and 39).

Comment 4:

Lines 210-211 “yet”; and line 371 “mediate” repeats twice. Please re-write.

Response:

All these sentences have been modified as suggested (page number 13, line numbers 212 and 213; page number 18, line numbers 401-403).

Additionally,

  1. The reference list has been modified as we have added several new references. Special attention is given to conform to the order of references and bibliographic style of the journal.
  2. The entire manuscript has been thoroughly checked and edited to ensure uniform style, organization, and quality.

On behalf of my co-authors, I once again express my sincere thanks to the erudite reviewer for the valuable suggestions and constructive input to improve the quality of our manuscript.

Round 2

Reviewer 1 Report

The authors have improved the manuscript adequately based on the comments and suggestions raised.

Reviewer 2 Report

I thank the authors for their kind reply and their efforts to revise the manuscript. Unfortunately, there are still very many important citations missing and the review is not focused enough. Unfortunately, I still think that the authors should take more time to focus on one aspect like Cyclin D.

Knowing that the schedule for the major revision is only 10 days. I would reject it in the present form with the option to resubmit it after significant and in-depth revision.